# How Does Breast Cancer Heterogeneity Determine Changes in Tumor Marker Levels in Saliva?

**DOI:** 10.3390/cimb47040216

**Published:** 2025-03-21

**Authors:** Elena I. Dyachenko, Lyudmila V. Bel’skaya

**Affiliations:** Biochemistry Research Laboratory, Omsk State Pedagogical University, 644099 Omsk, Russia; dyachenko.ea@gkpc.buzoo.ru

**Keywords:** breast cancer, saliva, MUC1, CEA, CA125, CA19-9, CYFRA 21-1, ferritin, CRP, epidermal growth factor receptors (HER2)

## Abstract

High heterogeneity of breast cancer is due to a large variety of cancer cell characteristics at the genomic, epigenomic, transcriptome, and proteomic levels. One of the difficulties is the separation of molecular biological subtypes based on the expression of tumor markers. Another problem is the difficulty of venipuncture in cancer patients when taking blood at different stages of patient care. Objectives: To identify statistically significant changes in the level of salivary tumor markers depending on the molecular biological subtype of breast cancer in order to improve understanding of the individual properties of each of its subtypes, 140 volunteers (breast cancer—110; healthy control—30) took part in the case–control study. Saliva was collected strictly before the start of treatment, and the content of ten tumor markers was determined by ELISA: EGFR2, CA15-3, CA27.29, MCA, CEA, CA125, CA19-9, CYFRA 21-1, ferritin, and CRP. The content of MUC1 antigens (CA15-3, CA27.29, and MCA) statistically significantly decreased in the luminal B(+) subtype of breast cancer. The CA19-9 antigen showed high sensitivity to low HER2 expression. A reliable increase in the level of CYFRA 21-1 in saliva was shown in luminal A and luminal B(-) breast cancer. The work demonstrates the diagnostic capabilities of saliva in measuring tumor markers in patients with breast cancer. It was also found that there are reliable differences in the expression level and set of tumor markers in saliva depending on the molecular biological subtype of breast cancer. Thus, CYFRA 21-1 significantly increases with luminal A and luminal B(-), but CRP only increases with luminal A. CA15-3, CA27.29, MCA, and CA19-9 significantly decrease with luminal B(+) breast cancer.

## 1. Introduction

Recently, the attention of researchers has been attracted by the possibility of determining tumor markers not as done traditionally in serum/plasma but in saliva [1,2]. Saliva is a promising biomaterial for clinical research due to its non-invasive collection method and ease of transportation and storage of biomaterial. Another distinctive feature of saliva is that it contains a wide range of various proteins [3,4]. It is known that CA125 can be a potential diagnostic marker for breast cancer in saliva, with acceptable sensitivity and specificity [5]. Elevated levels of CA15-3 have been found in the saliva of breast cancer patients [6], with a moderate correlation between serum and salivary CA15-3 levels [7]. Epidermal growth factor and CEA levels have been shown to be significantly elevated in the saliva of breast cancer patients [8]. Increased levels of CA15-3 and HER2 in saliva have also been found, positively correlating with serum in breast cancer patients [9]. The results of several studies indicate that the selected markers, especially when assessed as a diagnostic panel, have the potential to be used in initial detection (in combination with mammography and physical examination) and/or for follow-up monitoring as well as in comprehensive breast cancer screening [10].

However, to date, no comprehensive analysis of the diagnostic significance of salivary tumor markers for breast cancer has been conducted. In this study, we decided to measure the levels of MUC1 (CA15-3, CA27.29, MCA, and CA19-9) [11], MUC16 (CA125) [12], and CEA [13] in saliva as previously described breast cancer markers. According to our hypothesis, measuring CYFRA 21-1 and the tyrosine kinase human epidermal growth factor receptor 2 (EGFR2) should allow identifying breast cancer with positive and negative HER2 expression [14]. Ferritin and CRP are markers of inflammatory process activity [15,16]. At the systemic level, CRP reflects the presence of a systemic inflammatory process in the body, which can play an important role in the progression of breast cancer [17]. It is known that the increase in ferritin levels in breast cancer is proportional to the stage of the disease. The increase in ferritin levels is associated with the action of tumor necrosis factor, which is a variant of the cytoprotective response to suppress the oxidative stress reaction [18]. In this regard, we investigated these ten markers (EGFR2, CA15-3, CA27.29, MCA, CEA, CA125, CA19-9, CYFRA 21-1, ferritin, and CRP) in saliva with the aim of their possible use for the differential diagnosis of HER2-positive and HER2-negative subtypes of breast cancer.

We recently prepared a comprehensive review that systematizes all salivary indicators for breast cancer diagnostics [19]. It is important to note that most studies are characterized by small samples (mostly up to 50 patients), a lack of detailed description of the sample structure (stages and molecular biological subtype of breast cancer are not specified), and the study of single breast cancer markers in saliva. The aim of this study was to identify statistically significant changes in the level of salivary tumor markers depending on the molecular biological subtype of breast cancer in order to improve understanding of the individual properties of each of its subtypes.

## 2. Materials and Methods

### 2.1. Study Population

A total of 140 volunteers (breast cancer—110, aged 62.9 [38.2; 68.7] years; healthy controls—30, aged 46.1 [36.0; 58.0] years) participated in the case–control study. Inclusion into the groups was carried out in parallel. The inclusion criteria were female gender; Caucasian; age of 30–70 years; absence of any treatment at the time of the study, including surgical, chemotherapeutic, or radiation; absence of signs of active infection (including purulent processes); and oral cavity sanitation. Exclusion criteria included absence of histological verification of the diagnosis.

Patients with breast cancer were hospitalized for elective surgery or the first course of chemotherapy (Clinical Oncology Dispensary, Omsk, Russia). All patients in the main group had histologically and cytological confirmed invasive breast carcinoma. The distribution of patients by stages of breast cancer was as follows: stage IA + IB—19 (17.3%), stage IIA + IIB—43 (39.1%), stage IIIA + IIIB—25 (22.7%), and stage IIIC + IV—23 (20.9%). No breast pathologies were detected in the healthy volunteers as part of routine mammography.

Patients were recruited in such a way that the study group included an equal number of different molecular biological subtypes of breast cancer, namely luminal A (n = 22; 20%), luminal B(-) (n = 22; 20%), luminal B(+) (n = 22; 20%), non-luminal (n = 22; 20%), and triple-negative breast cancer (TNBC) (n = 22; 20%). Accordingly, 44 patients (40%) had positive expression of HER2 receptors, and 66 (60%) had negative expression. The following distribution was observed by stages: stage IA + IB—19 (17.3%), stage IIA + IIB—36 (32.7%), stage IIIA + IIIB—31 (28.2%), and stage IIIC + IV—24 (21.8%). In 47 (42.7%) patients, there was negative expression of estrogen receptors; in 63 (57.3%) patients, there was positive expression; in 59 (53.6%) patients, there was negative expression of progesterone receptors; in 51 (46.4%), there was positive expression. A low index of proliferative activity was noted in 34 (30.9%) patients and a high one in 76 (69.1%). According to the degree of differentiation, subgroups were identified: 46 patients with well and moderate differentiation (41.8%), 47 with poor differentiation (42.7%), and 17 (15.5%) patients for which there was no information on the degree of differentiation. A detailed description of the structure of the studied subgroups is given in Table 1.

### 2.2. Immunohistochemical Analysis

The Allred Scoring Guideline was used to assess the level of expression of estrogen receptors (ER), progesterone receptors (PR), and HER2 by IHC [20]. The level of expression of estrogen, progesterone, and HER2 receptors was assigned to one of four categories (0, 1+, 2+, and 3+) in accordance with the ASCO/CAP recommendations [21]. HER2 IHC results (1+) or 0 were interpreted as HER2-negative (HER2 without overexpression). If the HER2 status by IHC was indeterminate (2+), it was necessary to evaluate the presence of HER2 gene amplification by in situ hybridization (ISH) (Figure 1A). The result was assessed as the ratio of the number of copies of the HER2 gene (FISH) to the number of copies of chromosome 17 (CEP17). A sample in which FISH/CEP17 > 2 was considered HER2-positive. Patients with high levels of HER2 expression in the tumor (3+) were also considered HER2-positive. Ki-67 expression was determined as part of a standard breast cancer panel according to the manufacturer’s protocol [22]. The cut-off value for Ki-67 was defined as 20% (<20%—low Ki-67; >20%—high Ki-67).

Based on the expression level of ER, PR, and HER2 receptors and the Ki-67 proliferative activity index, breast cancer tumors were divided into luminal A, luminal B HER2-negative (luminal B(-)), luminal B HER2-positive (luminal B(+)), non-luminal HER2-positive (non-luminal), and TNBC (Figure 1B) [23].

### 2.3. Collection, Storage, and Pre-Treatment of Saliva Samples

Saliva samples were collected once at the hospitalization stage strictly before the start of treatment. Samples were collected in sterile polypropylene centrifuge tubes with a screw cap in a volume of 2 mL. Saliva samples were collected by spitting without additional stimulation in the range of 8–10 a.m., the time of maximum saliva secretion, on an empty stomach after preliminary rinsing of the mouth with water. Immediately after collection, the samples were centrifuged at 10,000× *g* for 10 min (CLn-16, Russia); then, 1 mL of the upper layer was collected, transferred to Eppendorf tubes, and stored at −80 °C until analysis.

### 2.4. Saliva Enzyme Immunoassay

EGFR2 (ng/mL) was determined by the sandwich enzyme immunoassay method (CEB867Hu, Cloud-Clone Corp., Houston, TX, USA). The content of tumor markers CA15-3 (catalog number K226, U/mL), CA27.29 (K227, U/mL), MCA (K228, U/mL), CEA (K224, ng/mL), CA125 (K222, U/mL), CA19-9 (K223, U/mL), and CYFRA 21-1 (K236, ng/mL) in saliva was determined by the solid-phase enzyme immunoassay method using the Hema kits (Moscow, Russia). The concentration of ferritin (T-8552, ng/mL) and CRP (A-9001, mU/mL) was determined by the highly sensitive enzyme immunoassay method using the Vector-Best kits (Novosibirsk, Russia). The analysis was performed on a Thermo Fisher Multiskan FC analyzer (Waltham, MA, USA). The concentration was calculated in accordance with the manufacturer’s instructions. The validation procedure for each test system included four analytical series. Each analytical series included the analysis of calibration standards, as well as the required number of quality control samples with a certain concentration of the corresponding tumor marker. Each sample was analyzed in two replicates.

### 2.5. Statistical Analysis

Statistical analysis was performed using Statistica 10.0 (StatSoft) software by a nonparametric method using the Mann–Whitney U-test. The distribution and homogeneity of variances in the groups were preliminarily checked. According to the Shapiro–Wilk test, the content of all the parameters being determined did not correspond to the normal distribution (*p* < 0.05). The conducted test for homogeneity of variances in the groups (Bartlett’s test) allowed us to reject the hypothesis that variances were homogeneous across the groups (*p* < 0.0001). Therefore, nonparametric statistical methods were used to process the data. The sample was described using the median (Me) and the interquartile range in the form of the 25th and 75th percentiles [LQ; UQ]. Differences were considered statistically significant at *p* ˂ 0.05.

## 3. Results

### 3.1. Salivary Tumor Marker Levels in Breast Cancer and Healthy Controls

When comparing the levels of tumor markers in the saliva of patients with breast cancer and healthy controls, it was found that the levels of only two markers changed statistically significantly: CA-125 and CYFRA 21-1 (Table 2). In this case, all markers can be divided into two subgroups by the change in the marker level in saliva. In the first subgroup, the marker levels in breast cancer will increase relative to the healthy control (EGFR2, CRP, CA-125, CEA, and CYFRA 21-1). In the second subgroup, the marker levels will decrease in breast cancer (CA15-3, CA 27.29, MCA, and CA19-9). For ferritin, the change in content in breast cancer can be considered insignificant.

Since the age of patients in the main and control groups differed quite significantly, a correlation analysis was preliminarily performed to check the relationship between the concentration of tumor markers in saliva and the age of volunteers. Only for CA-125 (*r* = 0.3122) and CEA (*r* = 0.2257) were weak correlations shown; in other cases, the correlations were statistically insignificant: EGFR2 (*r* = 0.0489), CRB (*r* = 0.0710), ferritin (*r* = −0.1385), CA15-3 (*r* = 0.0645), CA27.29 (*r* = 0.0186), MCA (*r* = 0.0676), CA19-9 (*r* = −0.1570), and CYFRA21-1 (*r* = −0.0067). For CA-125 and CEA, we calculated the concentration depending on the age group (up to 50 years and older). It was shown that in the group up to 50 years of age, the concentration of CA-125 and CEA was 290.0 [162.5; 409.0] U/mL and 83.26 [68.24; 89.28] ng/mL, whereas in the group aged greater than 50 years, the concentration was 365.8 [281.4; 459.9] U/mL and 88.57 [80.38; 95.15] ng/mL. When compared with healthy controls, it was shown that for CA-125, statistically significant differences persist for both subgroups regardless of age (*p* = 0.0457 and *p* = 0.0008 for the age up to 50 years and older, respectively). For CEA, differences were not shown without taking into account age and were not detected after additional separation of age groups.

### 3.2. Salivary Tumor Marker Levels Depending on Breast Cancer Stage

For a number of tumor markers, such as CRP, CYFRA 21-1, ferritin, CA27.29, and CA19-9, a sharp increase in content was observed at advanced stages of breast cancer compared to the healthy control (Figure 2A–D). For CA 27.29 and CA19-9, the change in content increased to such an extent that it reached values higher than in the control group (Figure 2C). However, a statistically significant increase was observed only for CA-125 at all stages of breast cancer (Figure 2A), CYFRA 21-1 at advanced stages (Figure 2B), and CA 19-9 at stage II breast cancer (Figure 2C).

### 3.3. Salivary Tumor Marker Levels Depending on the Expression of ER, PR, and HER2 Receptors; Proliferative Activity Marker Ki-67; and the Degree of Differentiation of Breast Cancer

At the next stage, we analyzed separately the effect of expression of estrogen, progesterone, and HER2 receptors as well as the proliferative activity index and the degree of tumor differentiation on the content of tumor markers in saliva. An interesting pattern of statistically significant increase in the CRP level in hormone receptor-positive breast cancer was shown (Table 3). Differences with the control group were significant in subgroups with high expression of estrogen and progesterone receptors for CRP and CYFRA 21-1 in the absence of expression-for EGFR2. In all cases, significant differences with the healthy control remained for CA-125 (Table 3).

An increase in the CRP level was also noted with high and medium degrees of tumor differentiation and a low proliferative activity index (Table 4). Increased levels of CA-125 and CYFRA 21-1 compared to healthy controls persisted in all cases.

The greatest differences were shown between breast cancer subgroups with different HER2 receptor expression (Figure 3A–C). Thus, with HER2-negative status, the levels of CRP, CYFRA 21-1 (*p* = 0.0028), and CA19-9 (*p* = 0.0047) increased, whereas with HER2-positive status, an increase in the levels of EGFR2 (*p* = 0.0222) and CA-125 (*p* < 0.0001) was observed as well as a decrease in the levels of CA15-3 (*p* = 0.0487), CA 27.29 (*p* = 0.0018), MCA (*p* = 0.0419), and CA19-9 (*p* = 0.0314).

### 3.4. Salivary Tumor Marker Levels Depending on the Molecular Biological Subtype of Breast Cancer

Thus, it seems possible to identify tumor markers, the change in which was characteristic only of HER2-positive subtypes of breast cancer (non-luminal and luminal B(+)), namely EGFR2, CA15-3, CA27.29, MCA, and CA19-9 (Table 5). For luminal HER2-negative subtypes, an increase in CYFRA 21-1 and CRP was characteristic. Regardless of HER2 expression, the level of CA-125 increased in breast cancer (Table 5). TNBC occupied an intermediate position: on the one hand, the level of CYFRA 21-1 increased for luminal subtypes, and on the other hand, the level of CA15-3 and MCA decreased for HER2-positive subtypes of breast cancer.

For CA19-9, an ambiguous pattern of change depending on HER2 expression was revealed, and a decrease in the content was noted in the luminal A subtype of breast cancer (Table 5). We analyzed the subgroup of patients with luminal A breast cancer and found that the subgroup was heterogeneous in its composition. Thus, seven patients (31.8%) had insignificant HER2 expression 1+ or 2+ according to the results of immunohistochemistry, while HER2 expression was not observed in the remaining patients. In the group of luminal B(-) breast cancer, there were only two patients with insignificant HER2 expression as well as in the TNBC subgroup. We identified this subgroup as a separate one and compared the content of tumor markers, the level of which significantly depended on the HER2 expression status (Figure 4).

It was shown that for all tumor markers, the concentrations in the HER2 low subgroup occupied an intermediate position between the subgroups with positive and negative HER2 expression (Figure 4). However, it was for CA19-9 that this subgroup was characterized by the lowest content compared to other subgroups. Thus, we noted that the level of CA19-9 in saliva responded even to minor changes in HER2 expression.

## 4. Discussion

Most serum components are known to be present in saliva [24]. Although the mechanisms of metabolite penetration from blood into saliva are theoretically clear, most studies have so far failed to adequately explain the mechanisms of tumor marker penetration into saliva. For example, the exact mechanism explaining the presence of CRP in saliva is unclear [25]. Although Ouellet-Morin and co-workers observed a moderate to strong relationship between saliva and serum CRP, it was shown that the correlation coefficients depend on the concentration of CRP and increase with increasing concentration [26]. CRP is known to be a key pro-inflammatory biomarker, acting as the main acute phase reactant and a biomarker of chronic low-grade inflammation, which is involved in carcinogenesis [27]. Our study showed a statistically significant increase in CRP levels with positive expression of hormones (estrogen and progesterone) as well as low proliferative activity and high tumor differentiation (Table 3 and Table 4). A significant increase in CRP levels was noted only in the luminal A subtype of breast cancer. Luminal A breast cancer refers to less aggressive subtypes of breast cancer. Recent studies have shown the crucial role of ferritin disturbances and the closely related disruption of the regulation of intracellular iron homeostasis; however, the molecular mechanisms underlying cancer-associated ferritin changes remain largely unknown and often lead to contradictory conclusions [28]. We have not shown any significant changes in salivary ferritin levels in breast cancer.

An interesting result of our work is that the groups of tumor markers CA 15-3, CA 27.29, MCA, and CA 19-9 showed a unidirectional decrease in concentration in the luminal B(+) type of breast cancer (Table 5).

It is known from literary sources that the content of CA19-9, CA15-3, CA27.29, and MCA increases in oncological diseases, including breast cancer [29]. In our study, we observed a statistically significant decrease in their levels in the saliva of patients with breast cancer, which reaches maximum values in the luminal B(+) type of breast cancer. To better understand such a multidirectional change in the content of tumor markers, it is necessary to turn to the nature of their origin. The listed antigens are different immunoreactive parts of one MUC1 molecule. Thus, CA15-3, CA27.29, and MCA are immunodominant regions of the MUC1 molecule [30]. Normally, such regions are hidden from the major histocompatibility complex I in a dense and branched N-glycosylated terminal domain. This masks the Toll-like receptor, which reduces the effector functions of T cells. Due to this, MUC1 has anti-inflammatory mechanisms. Tumor-associated MUC1 on the surface of the epithelial cell has a reduced density and length of the N-glycosylated terminal domain. Immunoreactive epitopes are exposed, which leads to stimulation of the immune system and activation of pro-inflammatory cytokines [31].

Regulation of non-aberrant MUC1 expression depends on the immune and hormonal systems. We suggest that the negative values of CA15-3, CA27.29, and MCA obtained by us among patients with breast cancer compared to the control are associated with damage to the epithelial cells of the oral cavity and salivary glands. Damage occurs because of activation of pro-inflammatory cytokines that have passed through the histohematological blood–saliva barrier (BSB). It is known that aberrant overexpression of the MUC1 and HER2 receptor is often observed in breast cancer [32]. Thus, markers CA15-3, CA27.29, and MCA can indirectly reflect changes occurring at the systemic level in breast cancer with HER2-positive expression, in our case, the luminal B(+) type of breast cancer.

The change in CA19-9 remains a controversial issue. The CA19-9 antigen is heterogeneous in its belonging to carrier proteins. According to studies, CA19-9 can be associated with such carrier proteins as MUC1, MUC5AC, MUC16, and ApoB. In this case, the CA19-9 antigen is expressed to the greatest extent on MUC1 [33]. In the study, we observed the lowest values of CA19-9 content in saliva in the groups of luminal A, luminal B(+), and non-luminal subtypes of breast cancer. In this regard, we cannot accurately classify CA19-9 as one of the groups with positive or negative HER2 expression.

It is known that the TNBC group is conditionally generalized. In studies of the TNBC gene cluster analysis, scientists have demonstrated a genetically determined division within TNBC into six subgroups: basal-like subtype 1 (BL1), basal-like subtype 2 (BL2), immunomodulatory (IM), mesenchymal (M), mesenchymal stem-like (MSL), and luminal androgen receptor (LAR) subtype [34]. In the same study, the majority of TNBC tumors within the LAR subtype were classified as luminal A or luminal B (82%), and none were classified as basal-like. This confirms the luminal origin of the LAR subtype. In this regard, there is an inaccurate classification of patients into molecular biological subtypes of breast cancer, where patients with the luminal A subtype fall into groups with TNBC. To date, there is no deep understanding of the characteristic features that divide the TNBC group into subgroups. We can only assume that the content of CA19-9 in saliva is close to normal among patients with TNBC due to its internal heterogeneity. At present, it is not appropriate to discuss the clinical significance and practical applicability of CA19-9 in patients with TNBC. Further studies and discussions are needed for this.

We showed in our experiment that some patients diagnosed with the luminal A subtype of breast cancer have low HER2 expression. As a result, we observed a statistically significantly low CA19-9 value in saliva in the group of luminal subtypes, characteristic of the MUC1 family antigens in the group of patients with the positive HER2 expression luminal B subtype of breast cancer. In our opinion, the direction in studying the biochemical and genetic features of breast cancer with low HER2 expression “+” or “+/−” is promising. It is known that the College of American Pathologists in 2023 released updates to the templates for reporting biomarker testing results in primary invasive breast carcinomas and in recurrent and metastatic tumors, which now include notes for low expression of the HER2 biomarker (HER2-low) [35].

CA125, also known as MUC16, was significantly increased in both positive and negative HER2 expression groups. Moreover, significant changes in its concentration can be observed in the group with positive HER2 expression. MUC16 (CA125) is expressed in human nasal mucosa and corneal epithelial cells, while the only sites of MUC1 synthesis in the oral cavity are the ducts of minor salivary glands [36]. Since the filtration of mucins through the BSB is difficult due to their large molecular weight (300–450 kDa) [37], we assume that changes in their concentration occur at the local level in saliva. In this case, pro-inflammatory cytokines TNF-α, INF-γ, and IL-1α that have passed through the BSB can act as regulators of CA125 overexpression. These cytokines influence NFκB, which leads to increased production of MUC16 (CA125) [38].

In our study, we observed a statistically significant increase in the CYFRA 21-1 content in saliva in the group of patients with luminal A and luminal B(-) subtypes of breast cancer. CYFRA 21-1 belongs to the soluble region of cytokeratin 19. The well-known function of cytokeratin 19 is the maintenance of cellular structure. Also, cytokeratin 19 can have an immunomodulatory effect due to the ability to transmit signals for the activation, development, and differentiation of B lymphocytes [39]. Cytokeratin 19 is highly expressed in luminal A subtype breast cancer and affects the proliferation of cancer cells. Increased expression of cytokeratin 19 affects the expression of the target gene E2F1, which is the main transcription factor for the transition of the cell to the S phase of the cell cycle. Cytokeratin 19 also activates the expression of cyclin-dependent kinases (CDKs) and cyclins, including D-type cyclins, which also activate the expression of the E2F1 gene. Excessive cytokeratin 19 activates caspase 3, which cleaves it. Thus, excess CYFRA 21-1 is formed in the blood. The molecular weight of CYFRA 21-1 is 30 kDa, which allows it to easily cross the BSB [40]. We suggest that the level of CYFRA 21-1 may directly correlate with the level in the blood.

CEA is a glycoprotein weighing 180 kDa and is one of the best-known tumor markers. Among patients with breast cancer, an increase in CEA correlates with the metastatic activity of the tumor and its relapse. CEA is often used in combination with other tumor markers to increase diagnostic sensitivity and specificity. The literature provides contradictory data on changes in CEA in saliva in breast cancer [41]. In a parallel comparison of the CEA level in blood plasma and saliva, it was shown that with a significant increase in CEA in plasma, no significant growth of CEA is observed in saliva. It should be noted that the content of CEA in saliva, even under normal conditions, is more than an order of magnitude higher than the content in blood plasma. The highest content of this protein is found in the salivary glands and their ducts. It can be assumed that CEA is produced by the tissues of the salivary glands, and its level in saliva does not correlate with the content in blood plasma. With a high background content of CEA in saliva, changes caused by the presence of oncological pathology do not cause its significant dynamics. In our study, we did not find any statistically reliable and significant changes, which allows us to conclude that its measurement in saliva in breast cancer is not advisable.

Increased expression of EGFR2 is present in all HER2-positive cancer types [42]. The EGFR family consists of EGFR itself, as well as EGFR2, EGFR3, and EGFR4 [43]. Active HER2 signaling promotes cell survival, growth, and dimerization of the receptor itself. There are two types of dimerization, ligand-dependent and ligand-independent dimerization. This depends on whether the receptor is a heterodimer or a homodimer [44]. HER2 is highly expressed in 20% of all breast cancer types [45]. We registered statistically significant changes in the EGFR2 level in saliva only in the luminal B(+) breast cancer subtype. Previously, it was proposed to use HER2 levels in saliva to identify HER2 types of breast cancer [46]. However, De Abreu Pereira and colleagues showed significant variations in HER2 levels in each subgroup, including the control group [47]. Laidi and colleagues also did not show a significant difference in clinical characteristics depending on positive and negative HER2 status [48]. Thus, the use of EGFR2 in saliva as a breast cancer marker is, in our opinion, questionable.

The limitations of this study are the lack of division of subjects into groups before and after menopause. This will be considered in future studies. The study had an insufficiently large sample size, which was limited primarily by the rarest non-luminal subtype of breast cancer, as well as statistically significant differences in age between the main and control groups. Despite the lack of correlation of the studied tumor markers with age, in our subsequent studies, we will try to reduce the age spread between the subgroups. We have shown an ambiguous trend in changes in the concentration of CA19-9, which does not coincide with changes in other tumor markers associated with MUC1. This issue certainly requires further clarification. The work did not consider the biological and technical variability of salivary biomarkers. This will be reflected in upcoming studies for a more in-depth and meaningful analysis of the behavior of tumor markers in saliva in breast cancer.

## 5. Conclusions

The results of the study show that the heterogeneity of breast cancer in this case is manifested in different levels of expression and a set of tumor markers in saliva depending on the molecular biological subtype. Thus, the level of CRP is statistically significantly higher in the luminal A subtype of breast cancer. The highest level is shown by CYFRA 21-1 in luminal A and luminal B(-) breast cancer. Tumor markers CA15-3, CA27.29, MCA, and CA19-9 statistically significantly decrease in HER2-positive subtypes of breast cancer (luminal B(+) and non-luminal). CA 125 does not show specificity with respect to molecular biological subtypes of breast cancer. The result of the study is evidence that saliva can be used as an informative diagnostic biomaterial in relation to CRP, CYFRA 21-1, CA15-3, CA27.29, MCA, and CA19-9 in patients with breast cancer. This is a pilot study in which we show for the first time a change in the content of tumor markers in saliva in patients with breast cancer.

## Figures and Tables

**Figure 1 cimb-47-00216-f001:**
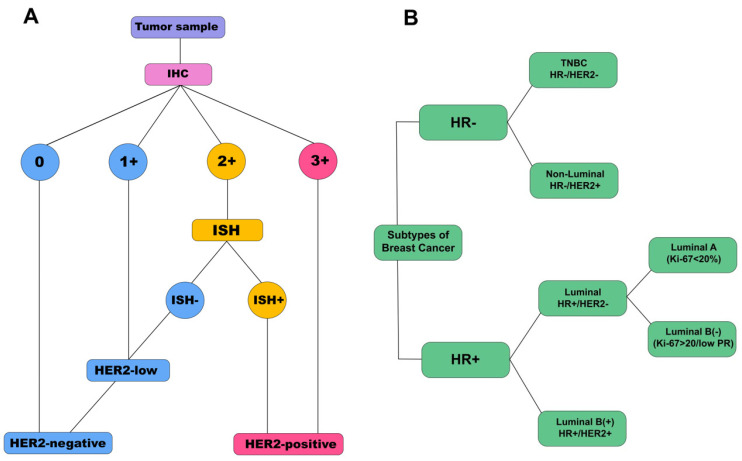
Schemes for determining the HER2 status of breast cancer (**A**) and determining the molecular biological subtype of breast cancer (**B**). HR—expression of hormonal receptors (estrogen and progesterone receptors); IHC—immunohistochemistry; ISH—in situ hybridization; TNBC—triple-negative breast cancer.

**Figure 2 cimb-47-00216-f002:**
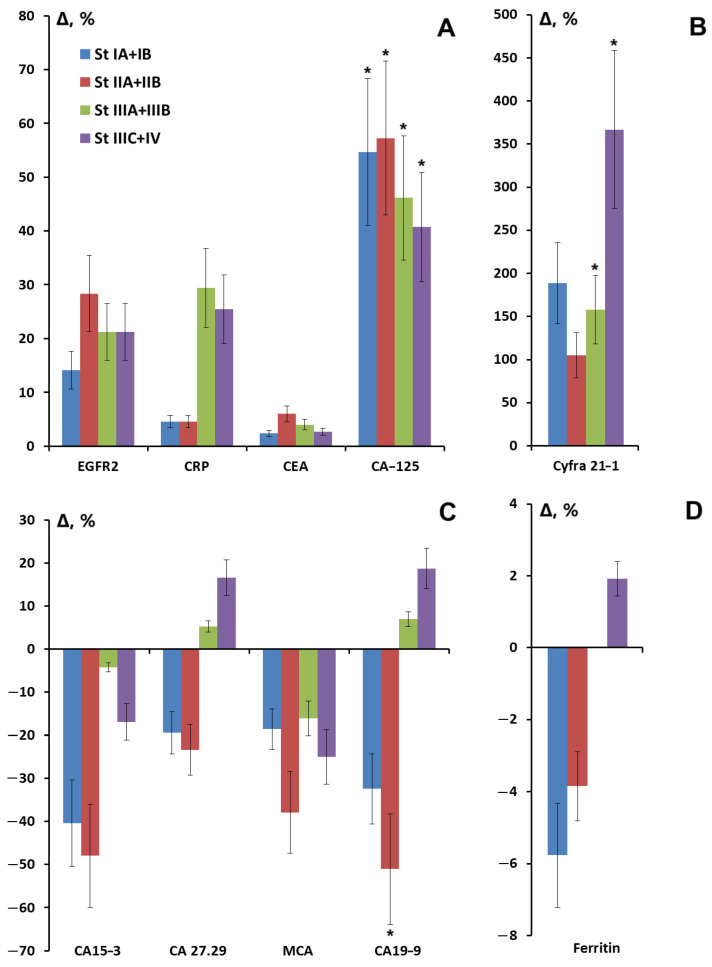
Relative change in the level of tumor markers in saliva depending on the stage of breast cancer: (**A**) EGFR2, CRP, CEA, and CA-125; (**B**) CYFRA 21-1; (**C**) CA15-3, Ca27.29, MCA, and CA19-9; (**D**) ferritin, %. *—differences with healthy controls are statistically significant, *p* < 0.05. Relative changes are calculated as the difference between the corresponding value for breast cancer and healthy controls relative to healthy controls (%).

**Figure 3 cimb-47-00216-f003:**
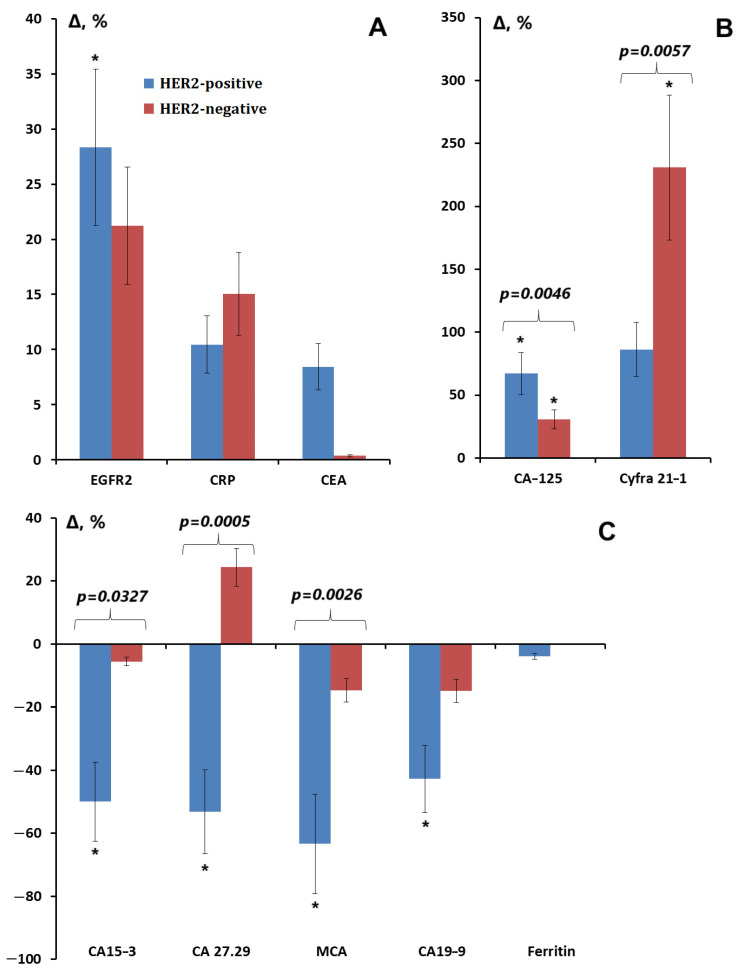
Relative change in the content of tumor markers in saliva depending on the expression of HER2 receptors: (**A**) EGFR2, CRP, and CEA; (**B**) CA-125 and CYFRA 21-1; (**C**) CA15-3, Ca27.29, MCA, CA19-9, and ferritin (%). *—differences with healthy controls are statistically significant, *p* < 0.05; *p*-values are given for statistically significant differences between subgroups.

**Figure 4 cimb-47-00216-f004:**
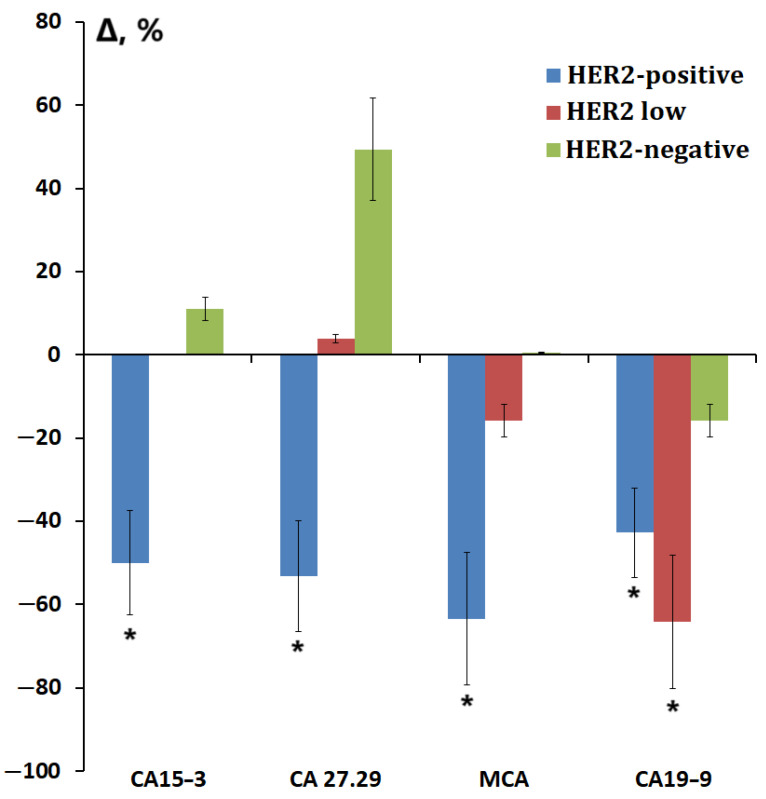
Relative change in the concentration of CA15-3, CA 27.29, MCA, and CA19-9 depending on HER2 expression, taking into account the HER2 low subgroup (%). *—differences with healthy controls are statistically significant, *p* < 0.05.

**Table 1 cimb-47-00216-t001:** Structure of breast cancer subgroups with different molecular biological subtypes.

Feature	Luminal A, n = 22	Luminal B(-), n = 22	Luminal B(+), n = 22	Non-Luminal, n = 22	TNBC, n = 22
Age, years	65.3 [51.7; 69.9]	64.5 [47.6; 66.6]	61.0 [48.2; 67.4]	57.6 [51.3; 68.5]	65.2 [41.6; 70.2]
Clinical Stage					
Stage IA + IB	22.7%	9.1%	13.6%	22.7%	18.2%
Stage IIA + IIB	31.8%	27.3%	36.4%	40.9%	27.3%
Stage IIIA + IIIB	18.2%	31.8%	31.8%	22.7%	36.3%
Stage IIIC + IV	27.3%	31.8%	18.2%	13.7%	18.2%
Lymph node status					
N0	36.4%	22.7%	59.1%	54.5%	40.9%
N1–3	63.6%	77.3%	40.9%	45.5%	59.1%
Degree of differentiation (G)					
G I + II	94.7%	38.9%	52.4%	25.0%	33.3%
G III	5.3%	61.1%	47.6%	75.0%	66.7%
Marker of proliferative activity Ki-67					
<20%	100%	13.6%	18.2%	4.5%	13.6%
>20%	0%	86.4%	81.8%	95.5%	86.4%

TNBC—triple-negative breast cancer.

**Table 2 cimb-47-00216-t002:** Salivary tumor marker levels in breast cancer and healthy controls.

Tumor Markers	Healhty Controls, n = 30	Breast Cancer, n = 110	*p*-Value
EGFR2, ng/mL	0.603 [0.474; 0.731]	0.731 [0.517; 0.944]	0.0685
CRP, mU/mL	0.153 [0.118; 0.212]	0.176 [0.122; 0.312]	0.2528
CEA, ng/mL	83.37 [68.14; 94.02]	86.94 [77.97; 94.32]	0.4262
CA125, U/mL	234.51 [100.83; 308.04]	350.98 [254.90; 448.69]	0.0001 *
CYFRA 21-1, ng/mL	2.04 [1.03; 7.13]	4.93 [1.87; 12.51]	0.0177 *
CA15-3, U/mL	39.6 [21.4; 92.7]	31.0 [14.1; 83.6]	0.2741
CA 27.29, U/mL	3.08 [2.11; 5.22]	2.66 [1.31; 6.47]	0.2373
MCA, U/mL	21.10 [6.24; 75.92]	16.61 [7.08; 47.11]	0.3967
CA19-9, U/mL	49.23 [21.64; 214.64]	37.59 [16.55; 111.91]	0.0963
Ferritin, ng/mL	15.6 [14.1; 18.2]	15.0 [14.1; 17.1]	0.8096

Values are given as median and interquartile range Me [25%; 75%]. * —the change in concentration between subgroups is statistically significant, *p* < 0.05.

**Table 3 cimb-47-00216-t003:** Salivary tumor marker levels in breast cancer depending on estrogen and progesterone receptor expression.

Tumor Markers	Estrogen Receptors		Progesterone Receptors	
	ER(-), n = 47 (Group 1)	ER(+), n = 63 (Group 2)	PR(-), n = 59 (Group 1)	PR(+), n = 51 (Group 2)
EGFR2, ng/mL	0.731 [0.560; 0.902]	0.774 [0.474; 1.030]	0.731 [0.560; 0.902]	0.731 [0.474; 1.115]
	-	p2-HC = 0.0439	p1-HC = 0.0303	-
CRP, mU/mL	0.135 [0.111; 0.193]	0.224 [0.148; 0.406]	0.136 [0.120; 0.218]	0.215 [0.150; 0.374]
	p1-2 = 0.0004	p1-2 = 0.0004, p2-HC = 0.0274	p1-2 = 0.0040	p1-2 = 0.0040
CEA, ng/mL	84.05 [75.30; 92.60]	87.18 [79.39; 95.33]	88.65 [77.75; 93.86]	85.60 [78.22; 95.00]
CA125, U/mL	385.49 [229.61; 472.16]	330.59 [258.63; 421.18]	385.4 [234.31; 460.59]	330.59 [254.90; 410.20]
	p1-HC = 0.0002	p2-HC = 0.0004	p1-HC = 0.0001	p2-HC = 0.0014
CYFRA 21-1, ng/mL	4.22 [1.77; 9.52]	5.28 [2.45; 15.48]	4.22 [1.56; 9.88]	5.52 [2.89; 14.75]
	-	p2-HC = 0.0085	-	p2-HC = 0.0064
CA15-3, U/mL	25.11 [13.32; 83.65]	36.94 [14.95; 84.69]	25.11 [12.96; 73.75]	37.86 [18.22; 84.79]
CA 27.29, U/mL	1.94 [1.21; 5.09]	2.99 [1.40; 6.56]	2.18 [1.07; 5.09]	3.06 [1.54; 6.72]
MCA, U/mL	12.99 [5.67; 31.32]	17.71 [8.01; 56.73]	12.99 [5.67; 31.32]	17.78 [8.64; 71.97]
CA19-9, U/mL	41.64 [16.64; 111.91]	32.55 [16.18; 125.27]	42.18 [16.27; 116.91]	28.73 [16.91; 108.91]
Ferritin, ng/mL	15.6 [14.7; 17.5]	15.0 [14.1; 16.8]	15.6 [14.7; 16.8]	14.9 [13.8; 17.1]

The HC index indicates statistically significant differences with healthy controls, *p* < 0.05; 1 and 2 are indices denoting subgroups with negative and positive expression of estrogen and progesterone receptors.

**Table 4 cimb-47-00216-t004:** Salivary tumor marker levels in breast cancer depending on the proliferative activity index and tumor differentiation degree.

Tumor Markers	Degree of Differentiation		Proliferative Activity Index	
	GI + II, n = 46 (Group 1)	GIII, n = 47 (Group 2)	Ki-67 Low, n = 32 (Group 1)	Ki-67 High, n = 85 (Group 2)
EGFR2, ng/mL	0.731 [0.517; 1.030]	0.731 [0.517; 0.902]	0.795 [0.496; 1.030]	0.731 [0.560; 0.902]
	-	-	p1-HC = 0.0407	-
CRP, mU/mL	0.206 [0.122; 0.374]	0.150 [0.118; 0.218]	0.206 [0.144; 0.390]	0.160 [0.120; 0.258]
	p1-2 = 0.0380	p1-2 = 0.0380	-	-
CEA, ng/mL	89.34 [81.65; 97.13]	84.38 [77.97; 91.68]	88.97 [77.71; 96.04]	86.69 [77.97; 93.07]
CA125, U/mL	350.29 [258.24; 504.12]	350.39 [234.31; 421.18]	346.27 [278.92; 454.64]	350.39 [234.31; 448.63]
	p1-HC = 0.0005	p2-HC = 0.0005	p1-HC = 0.0007	p2-HC = 0.0003
CYFRA 21-1, ng/mL	5.26 [1.87; 14.36]	4.53 [1.56; 9.39]	4.58 [2.12; 11.69]	5.07 [1.86; 12.86]
	p1-HC = 0.0142	-	p1-HC = 0.0304	p2-HC = 0.0262
CA15-3, U/mL	31.36 [14.95; 75.83]	20.62 [13.32; 86.67]	33.27 [17.15; 66.09]	29.19 [13.78; 84.69]
CA 27.29, U/mL	2.88 [1.31; 7.27]	2.58 [1.33; 5.56]	2.66 [1.36; 6.64]	3.06 [1.32; 6.47]
MCA, U/mL	13.15 [7.34; 51.00]	17.78 [7.04; 44.24]	16.91 [8.61; 72.32]	16.48 [6.61; 35.27]
CA19-9, U/mL	35.59 [16.27; 108.91]	40.64 [21.36; 116.91]	23.73 [15.14; 97.32]	42.36 [16.91; 116.91]
	-	-	p1-HC = 0.0323	-
Ferritin, ng/mL	14.7 [13.8; 16.8]	15.6 [15.0; 17.9]	15.0 [14.0; 16.9]	15.4 [14.4; 17.1]
	p1-2 = 0.0394	p1-2 = 0.0394	-	-

The HC index indicates statistically significant differences with healthy controls, *p* < 0.05; 1 is the index denoting the subgroups GI + II and Ki-67 low; 2 is the index denoting the subgroups GIII and Ki-67 high.

**Table 5 cimb-47-00216-t005:** Relative change in levels of tumor markers in saliva in different molecular biological subtypes of breast cancer, %.

Tumor Markers	HER2-Positive		HER2-Negative		
	Non-Luminal, n = 22	Luminal B(+), n = 22	Luminal A, n = 22	Luminal B(-), n = 22	TNBC, n = 22
EGFR2, ng/mL	25 * *p* = 0.0131	32	11	25	14
CRP, mU/mL	−18	35	137 * *p* = 0.0027	5	−7
CEA, ng/mL	8	8	5	3	−4
CA125, U/mL	71 * *p* < 0.0001	59 * *p* = 0.0014	44 * *p* = 0.0080	24 * *p* = 0.0184	15
CYFRA 21-1, ng/mL	98	56	328 * *p* = 0.0198	329 * *p* = 0.0249	183
CA15-3, U/mL	−36	−65 * *p* = 0.0382	19	−4	−37
CA 27.29, U/mL	−57 * *p* = 0.0016	−25 * *p* = 0.0348	41	10	19
MCA, U/mL	−60	−65 * *p* = 0.0301	54	−12	−36
CA19-9, U/mL	−34	−55 * *p* = 0.0384	−52 * *p* = 0.0324	−4	35
Ferritin, ng/mL	−3	−4	−6	−3	2

Blue color shows an increase in concentration; red color shows a decrease in the concentration of the tumor marker. The more intense the color, the more significant the changes. *—the differences with healthy controls are statistically significant.

## Data Availability

The raw data supporting the conclusions of this article will be made available by the authors on request.

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
