# Peer review of "How Does Breast Cancer Heterogeneity Determine Changes in Tumor Marker Levels in Saliva?"

_cimb, 2025, doi:10.3390/cimb47040216_

Round 1
Reviewer 1 Report
Comments and Suggestions for Authors
This study by Dyachenko et al. investigates the influence of breast cancer subtypes on tumor marker levels in saliva. The findings, particularly the variations in CYFRA 21-1 and CA15-3 across different subtypes, suggest that saliva-based diagnostics could serve as a complementary tool to existing methods. Below are my suggestions to enhance the scientific rigor and clarity of the manuscript:
-
The authors state that no comprehensive assessment of salivary breast cancer tumor markers exists. However, a systematic review by Porto-Mascarenhas et al. (PMID: 28109406) has already evaluated relevant studies. It would be beneficial to clearly define the novelty of this work in the introduction and discussion sections.
-
The inclusion and exclusion criteria should be explicitly stated. Additionally, demographic and pathological details of the participants should be presented in a table within the methods section for clarity.
-
To improve reproducibility, it is essential to provide the specific kit number, manufacturer details, and protocol for the Hema kits (Russia) used in the study.
-
Table 2 requires formatting improvements. Specifically, the 95% confidence intervals and statistical results should be clearly presented and easy to interpret.
-
Figures, particularly Figures 3 and 4, should maintain consistent formatting, including uniform color schemes, font sizes, and the inclusion of statistical markers such as asterisks or p-values for clarity.
-
The cut-off values and clinical thresholds for Ki67, PR+/-, ER+/-, and HER2 expression should be explicitly mentioned in either the methods or results section.
-
The clinical relevance of the findings warrants further discussion, especially regarding their potential application in triple-negative breast cancer and immune stratification.
-
The conclusion is currently too long and should be condensed to highlight the key findings without unnecessary repetition.
-
Addressing the biological and technical variability of salivary biomarkers would provide a more thorough and insightful discussion.
- The manuscript would benefit from language editing to improve readability, clarity, and scientific precision.
Reviewer 2 Report
Comments and Suggestions for Authors
The authors present a pilot case-control study aimed at evaluating the diagnostic potential of saliva by comparing the levels of ten tumor markers between breast cancer patients and healthy controls. In this study, 110 breast cancer patients (stratified by molecular subtypes such as luminal A, luminal B [both HER2-positive and HER2-negative], non-luminal, and triple-negative) and 30 healthy volunteers were used. Saliva samples were collected before treatment. The researchers measured ten tumor markers (EGFR2, CA15-3, CA27.29, MCA, CEA, CA125, CA19-9, CYFRA 21-1, ferritin, and CRP) using ELISA. The study identifies statistically significant changes in specific markers depending on the molecular biological subtype of breast cancer. The study concludes that saliva may serve as an informative diagnostic biofluid for breast cancer, with specific marker profiles varying by molecular subtype.
Major comments:
- The title does not fully reflect the content of the study. A more precise and descriptive title would be preferable.
- At the beginning of the introduction, authors should consider restructuring the text to first provide a brief background on the justification for using saliva as an attractive medium (e.g., its non-invasive collection, reduced patient discomfort, etc.) before delving into specific markers. (Line 35)
- In lines 152-161, the text reads more like a discussion than an objective explanation of the results. Consider moving this section to the Discussion, where interpretative comments are more appropriate.
- There are results presented in graphs (Figures 2 and 3) as well as in tables (Tables 2 and 3). Why? Since all the data is non-parametric, should the graphs not be presented as boxplots with medians and interquartile ranges? In Figure 3, the color legend should appear in the first graph, as done in Figure 2.
- In Table 3, it is difficult to identify the p-values and determine which are significant. What does p1-2, or p2-HC mean? Additionally, a * is mentioned in the footnote but does not appear in the table. Please clarify the table.
- Please also improve table 6.
- The discussion section of the manuscript is too extensive and should be improved to be more concise and objective.
- Future perspectives should be more clearly defined and objective. The authors did not separate patients into pre- and post-menopausal groups, and this limitation should be emphasized in future perspectives.
Minor comments: - Please remove the words “Objectives,” “Methods,” “Results,” and “Conclusions.” from abstracts.
- The statistical software used should be removed from the abstract as it is not relevant.
- The journal guidelines allow a maximum of 10 keywords. Please reduce the number accordingly.
- In the Materials and Methods section, specify that subtype detection was performed by investigating specific biomarkers using IHC.
- In Figure 1, “Ki-67” appears without a hyphen, while in the text it is written with a hyphen. Please ensure consistency by adding the hyphen in Figure 1.
- In Figure 1, remove the abbreviations "ER" and "PR" from the legend, as they do not appear in the image.
- Please add explanations for "ISH" and "IHC" in Figure 1’s caption.
- The colors of “IHC,” “2+,” and “ISH” in Figure 1(a) should not be the same as the text box colors in Figure 1(b).
- The labels “(a)” and “(b)” in Figure 1 should appear at the top of the image rather than at the bottom, as in the other figures. To maintain consistency, the labels should be “A” and “B,” as used in the other figures.
- Remove the word "Note" from the legend of all Tables.
- Divide the Materials and Methods section into subsections, such as "Study Population," "IHC”, “ELISA," and "Statistical Analysis," for better readability.
- The authors should cite the reference used to classify the breast cancer subtypes.
- It would be desirable to specify the geographical location and the hospitals in which the samples were taken.
- Line 38, please add space between “salivary” and “CA15-3 levels”.
- The third sentence of the discussion is too long and unclear. Please rephrase for clarity.
- In line 241, CEP17<2 (the control used in FISH for HER2) is mentioned, but this abbreviation only appears here. To clarify for the reader, this indication of the probe and the ratio should be included in the Materials and Methods section.
- Line 428, please add “De Abreu Pereira and colleagues”.
- In line 270, replace "et al." with "and co-workers."
- In line 429, replace "et al." with "and colleagues."
- In Table 5, a color legend would be desirable.
- Please do not mention Figures 3 and 4 in the legend of Figure 2.
- The terminology used for molecular subtypes (e.g., Luminal A, Luminal B(-), Luminal B(+), non-luminal, TNBC) should be used consistently throughout the manuscript.
The English language in the manuscript requires improvement, as some sentences are quite long and could benefit from clearer structure and phrasing.
Round 2
Reviewer 1 Report
Comments and Suggestions for Authors
-
Reviewer 2 Report
Comments and Suggestions for Authors
The authors have addressed the suggestions, leading to improvements and greater clarity in the manuscript.
However, before publication, there are a few minor considerations to consider. The authors have added more information regarding the inclusion of patients in the study, but I believe there is unnecessary overlap between the information in lines 75-80 and lines 81-82. Please review and adjust this section accordingly.
After these revisions, I believe the manuscript will be ready for publication.
Comments on the Quality of English LanguageNot applicable
